# Inflammatory Skin-Derived Cytokines Accelerate Osteoporosis in Mice with Persistent Skin Inflammation

**DOI:** 10.3390/ijms21103620

**Published:** 2020-05-20

**Authors:** Kento Mizutani, Kana Isono, Yoshiaki Matsushima, Karin Okada, Ai Umaoka, Shohei Iida, Koji Habe, Kohei Hagimori, Hidetoshi Yamazaki, Keiichi Yamanaka

**Affiliations:** 1Department of Dermatology, Mie University Graduate School of Medicine, 2-174 Edobashi, Tsu, Mie 514-8507, Japan; k-mizutani@med.mie-u.ac.jp (K.M.); matsushima-y@clin.medic.mie-u.ac.jp (Y.M.); okadakarin@clin.medic.mie-u.ac.jp (K.O.); umaokaai@clin.medic.mie-u.ac.jp (A.U.); kmcasters@clin.medic.mie-u.ac.jp (S.I.); habe-k@clin.medic.mie-u.ac.jp (K.H.); 2Stem Cell and Developmental Biology, Mie University Graduate School of Medicine, 2-174 Edobashi, Tsu, Mie 514-8507, Japan; k-isono@doc.medic.mie-u.ac.jp (K.I.); yamazaki@doc.medic.mie-u.ac.jp (H.Y.); 3Medicines Development Unit Japan, Eli Lilly Japan K.K., 5-1-28 Isogamidori, Chuo-ku, Kobe, Hyogo 651-0086, Japan; hagimori_kohei@lilly.com

**Keywords:** inflammatory skin mouse model, psoriasis, osteoporosis, minodronate, anti-RANKL antibody

## Abstract

Secondary osteoporosis can also be caused by chronic inflammatory skin disease as well as rheumatoid arthritis or inflammatory bowel disease. However, the exact role of osteoporosis in inflammatory skin conditions has not been elucidated. Using a mouse model of dermatitis, we investigated the pathophysiology of osteoporosis in inflammatory skin conditions and the therapeutic impact of osteoporosis medication on inflammatory skin disease. We employed model mice of spontaneous skin inflammation, specifically overexpressing human caspase-1 in the epidermis. Bone density and the expression of various mRNAs in the femur were examined by micro CT and RT-PCR. The effects of minodronate and anti-RANKL antibody on bone structure, histology, and femur blood flow were studied. The mouse model of skin inflammation showed a marked decrease in bone density compared to wild-type littermates with abnormalities in both bone resorption and formation. Minodronate improved bone density by decreasing osteoclasts, but anti-RANKL antibody did not improve. In the dermatitis model, the blood flow in the bone marrow was decreased, and minodronate restored this parameter. A model of persistent dermatitis exhibited marked osteoporosis, but the impact of chronic dermatitis on osteoporosis has not been thoroughly investigated. We should explore the pathogenesis of osteoporosis in skin inflammatory diseases.

## 1. Introduction

Osteoporosis is one of the most common diseases and a critical public health issue. Since osteoporosis causes patients to be bedridden due to bone fracture, prevention is important, not only to improve their quality of life, but also for economic issues [1]. Osteoporosis is characterized by a systemic impairment of bone mass and microarchitecture, which causes fragility fractures [2]. Osteoporosis is caused by an imbalance between bone formation and destruction, which are processes regulated by osteoblasts and osteoclasts, respectively. Previous studies have reported that lowered bone mineral density (BMD) is associated with the incidence of chronic inflammatory diseases, such as rheumatoid arthritis (RA), inflammatory bowel disease (IBD), and systemic lupus erythematosus [3,4,5]. These chronic inflammatory diseases are associated with the production of pro-osteoclastogenic cytokines, such as tumor necrosis α (TNF-α), interleukin-6 (IL-6), and receptor activator of NF-*κ*B ligand (RANKL), which activate osteoclasts [6].

Recent meta-analyses reported that, similarly to RA or IBD, atopic dermatitis (AD) and psoriasis vulgaris (Pso) are associated with osteoporosis [7,8,9,10]. AD and Pso are chronic and treatable inflammatory skin diseases. Osteoporosis drugs are divided into antiresorptive and anabolic agents, and are administered depending on the cause and condition of osteoporosis [11]. We have previously proven that the overproduction of skin-derived inflammatory cytokines results in organ failure, as is the case with cardiovascular and cerebrovascular disorders [12,13]. We reasoned that inflammatory cytokines from persistent skin lesions could also impact the bone. However, there is no report which was shown to observe secondary osteoporosis in a skin inflammatory mouse model. KCASP1Tg is a transgenic mouse that we have generated by ligating the human caspase-1 to the keratin 14 promoter, and it develops persistent dermatitis with interleukin (IL)-1β and various inflammatory cytokines secretion from inflamed skin [14]. Since secondary osteoporosis was observed in our spontaneous dermatitis mouse model, we investigated the association between long-lasting dermatitis and osteoporosis, and explored the effectiveness of osteoporosis medication.

## 2. Results

### 2.1. KCASP1Tg Mice EXHIBIT Osteoporosis

KCASP1Tg mice began to develop facial skin symptoms around 8 weeks of age, subsequently spreading to the whole body. BMD was screened at the distal femur metaphysis. BMD of KCASP1Tg mice decreased significantly compared to wild-type littermates at 12 weeks of age (Figure 1A). The cortical width of the femur was significantly thinner in KCASP1Tg mice, as assessed by histological hematoxylin eosin (HE) staining (Figure 1B,C). The level of the sensitive marker for bone resorption, serum tartrate-resistant acid phosphatase 5b (TRACP-5b), was significantly increased in KCASP1Tg mice, while the level of the biomarker for bone formation, serum osteocalcin, was not affected (Figure 1D,E). The mRNA was extracted from the femur and real-time polymerase chain reaction (RT-PCR) was performed. We measured the mRNA expression of genes involved in bone formation, such as alkaline phosphatase (*Alpl*), collagen type I (*Col1a1*), and bone gamma-carboxyglutamate (*Bglap*, osteocalcin). *Alpl* is an enzyme related to bone formation activity, *Col1a1* and *Bglap* a bone matrix that forms the bone skeleton, which were produced from osteoblasts. *Alpl*, *Col1a1*, and *Bglap* expression was significantly decreased in KCASP1Tg mice (Figure 1F–H). On the other hand, the mRNA levels of members of the tumor necrosis factor superfamily (*Tnfsf11*, *RANKL*), cytokines produced by osteoblasts promoting osteoclast differentiation and activation, and of vascular endothelial growth factor-a (*Vegfa*), which activates mature osteoclasts, did not differ between KCASP1Tg and wild-type mice (Figure 1I,J).

Next, we examined the expression of inflammatory cytokines in the femur. TNF*-α* expression was significantly increased in KCASP1Tg mice, and *IL-1α* was significantly decreased (Figure 2A,C). On the other hand, there was no change in the expression of IL-6, IL-1*β* or IL-23*a*, which are cytokines associated with osteoporosis (Figure 2B,D,F). IL-17A expression was undetectable (Figure 2E).

### 2.2. Analysis of Trabecular and Cortical Bone Structure by µCT: The Effect of Minodronate and Anti-RANKL Antibody on Bone Structure

We treated osteoporosis in KCASP1Tg mice by the subcutaneous administration of minodronate and OYC1 (anti-RANKL antibody) to mice from 6 to 16 weeks of age, and analyzed the bone structure by micro computed tomography (*µ*CT). The analysis of femurs confirmed the decrease of BMD and a marked augmentation of bone mass in minodronate-treated KCASP1Tg mice (Figure 3A,B). The bone volume per tissue volume (BV/TV) in trabecular bone is the volume of mineralized bone per unit volume. This index was reduced by 52% in KCASP1Tg mice compared to wild-type littermates (Figure 3C). In addition, the trabecular number (Tb.N, Figure 3D) and trabecular thickness (Tb.Th, Figure 3F) were significantly decreased in KCASP1Tg mice, but the trabecular separation (the distance between the trabeculae, Tb.Sp) was increased (Figure 3E). Minodronate significantly improved BMD, BV/TV, Tb.N, Tb.Sp, and Tb.Th, while OYC1 only produced a slight improvement.

KCASP1Tg mice also showed the exacerbation of cortical bone resorption compared to wild-type mice (Figure 4A). The cortical bone volume per all bone volume (Cv/Av) showed a 32% reduction, and both cortical thickness (Ct) and cortical bone section area (CS) were significantly decreased in KCASP1Tg compared to wild-type mice (Figure 4B–D). Porosity is the proportion of lumen in cortical bones and increases in osteoporosis. In KCASP1Tg mice, the porosity was inconspicuous in the diaphyseal part, and decreased at the distal femur metaphysis, albeit the difference was not significant (Figure 4E). In cortical bone, minodronate improved the porosity, but OYC1 did not result in any improvement. The data indicated that chronic dermatitis affected trabecular and cortical bone formation, but could be ameliorated by bisphosphonates.

### 2.3. Histological Analysis Revealed Osteoclast Activation in KCASP1Tg Mice

To evaluate the cellular mechanisms underlying the changes in bone parameters, a static histomorphometry of femoral diaphysis was performed in 16-week-old mice. The number of osteoblasts per bone surface (Ob.N/BS) and the eroded surface per bone surface (ES/BS) were measured by HE staining (Figure 5A). Ob.N/BS was significantly decreased and ES/BS was increased in KCASP1Tg mice compared to wild-type littermates (Figure 5B,C). ES/BS was not improved in both minodronate- and OYC1-treated KCASP1Tg mice. Next, we stained the osteoclasts using the osteoclast marker, tartrate-resistant acid phosphatase (TRAP) (Figure 5D). Osteoclast number per bone surface (Oc.N/BS) and osteoclast surface per bone surface (Oc.S/BS) were significantly increased in KCASP1Tg mice compared to wild-type mice (Figure 5E,F). Minodronate treatment reduced both Oc.N/BS and Oc.S/BS, whereas OYC1 only ameliorated Oc.S/BS.

### 2.4. Femoral Blood Flow was Decreased in KCASP1Tg Mice, but Improved after Minodronate Administration

Finally, we evaluated the blood flow in the femur by fluorescent staining using the vascular endothelial cell marker, panendothelial cell antigen antibody (MECA-32). In KCASP1Tg mice, the diameter of the blood vessels in the bone marrow was narrowed, and the proportion of the area stained by MECA-32 was lower compared to wild-type littermates (Figure 6A,B). Interestingly, minodronate improved the vascular area.

## 3. Discussion

In the current study, we proved that mice with skin eruption develop osteoporosis. The process of bone remodeling depends on the balance between bone resorption by osteoclasts and bone formation by osteoblasts. Chronic inflammatory diseases such as RA, IBD, and inflammatory skin disease induce secondary osteoporosis. Therefore, we hypothesized that inflammatory cytokines could adversely affect bone remodeling. The serum levels of TRACP-5b and osteocalcin have been identified as markers of bone resorption and bone formation, respectively [15]. In the present study, BMD was significantly decreased and the serum level of TRACP-5b was significantly increased in KCASP1Tg mice compared to wild-type mice, and it suggested that dermatitis model mice might cause osteoporosis with excessive bone resorption. The femur mRNA levels of genes involved in bone formation, including *Alpl*, *Col1a1*, and *Bglap*, were significantly downregulated; we speculated that serum osteocalcin would be decreased in KCASP1Tg mice. One characteristic complication in adult KCASP1Tg mice is renal dysfunction, which leads to the decreased excretion of osteocalcin, possibly explaining the similar level of serum osteocalcin observed in KCASP1Tg and wild-type mice [13,16]. On the other hand, the inflammatory cytokine, TNF-α, was significantly increased in the femur of KCASP1Tg mice. Interestingly, IL-1*β* and IL-6, which are upregulated in the skin of KCASP1Tg mice (data not shown), were at similar levels in the femur of KCASP1Tg and wild-type mice. Besides, RANKL (*Tnfsf11*), which is activated by inflammatory cytokines, was not overexpressed in the model mice. Osteoblasts express RANKL upon stimulation with TNF-*α*, IL-1 or IL-6, and bind to the receptor activator of NF-*κ*B (RANK), expressed on osteoclast precursor cells, inducing osteoclast differentiation [17]. The lack of increase in *RANKL* expression, despite the increase in TNF*-α* expression, is puzzling. However, TNF-*α* itself acts as an inhibitor of osteoblast differentiation from osteoclast precursor cells, thus reducing the proportion of osteoblasts [18].

In order to characterize the effects of chronic dermatitis and the impact of osteoporosis drugs on bone microarchitecture, *µ*CT was used to measure both trabecular and cortical parameters. Our results suggested that the persistence of dermatitis affected not only trabecular but also cortical bone structure. In patients with psoriatic arthritis (PsA), BMD, trabecular thickness, and trabecular number are significantly decreased, and the duration of psoriatic dermatitis is associated with trabecular bone loss [19]. Therefore, inflammatory cytokines produced by psoriatic skin might disrupt bone homeostasis. In our dermatitis model, deteriorated BMD, trabecular and cortical bone structure were significantly improved by the administration of minodronate. In addition, osteoclast was significantly suppressed by minodronate. However, anti-RANKL antibody did not improve BMD and osteoclast number, despite administration of a sufficient dose. Osteoclasts are differentiated by RANKL, but under inflammatory conditions, the differentiation may be promoted by a RANKL-independent pathway, which is mediated by inflammatory cytokines such as TNF-*α* and IL-6 [20,21,22]. On the other hand, when human peripheral blood CD14 + monocytes are stimulated and cultured with monocyte colony stimulating factor and IL-17, osteoclasts differentiate in a concentration-dependent manner [23]. Recent studies have shown that the use of cytokine inhibitor improves BMD in PsA patients [24]. In our model, a large amount of inflammatory cytokines are produced from the skin lesions (manuscript in preparation); therefore, we tried to prevent osteoporosis using antibodies against inflammatory cytokines, such as TNF-*α*, IL-1*α/β*, and IL-17A/F, but BMD did not change in any of the groups (data not shown). This could be due to several reasons. One explanation could be related to the utilized dose of antibody. Specifically, in each injection, we administrated 10 μg/body of anti-mouse TNF-*α*, IL-1*α/β* or IL-17A/F antibody (BioLegend) intraperitoneally into KCASP1Tg mice, three times a week, for 10 weeks, from week 6 to week 16, based on past our report [11]. Due to the high severity of dermatitis, a higher amount of antibody therapy may be required, because skin symptoms also did not improve by these cytokine therapies (data not shown). For identifying the relationship of cytokines and osteoporosis in the model mouse, additional research will be required. Alternatively, other factors may be responsible for the accelerated osteoporosis.

We previously reported that persistent dermatitis may result in cardiovascular disorders, cerebrovascular disorder, and lower limb blood flow impairment in KCASP1Tg mice [11,12]. It has also been reported that impaired blood flow causes decreased angiogenesis and bone formation in aged mice, and that blood flow is enhanced by the administration of bisphosphonate [25]. We confirmed the suppression of femoral blood flow in the mouse model of chronic dermatitis, while bisphosphonate administration improved the blood flow, as previously reported. Lower limb blood flow obstruction is associated with chronic inflammation, and may result in bone loss.

In conclusion, persistent dermatitis might cause osteoporosis, affecting both bone resorption and formation. To date, the impact of chronic dermatitis on osteoporosis has not been thoroughly investigated. We believe that the clarification of this link will help elucidate the pathogenesis of osteoporosis in skin inflammatory diseases.

## 4. Materials and Methods

### 4.1. Mouse Study

Female transgenic mice (5–52 weeks old) in which keratinocytes specifically overexpress the human caspase-1 gene under the keratin 14 promoter (KCASP1Tg) and C57BL/6N littermates mice were used. The experimental protocol was approved by the Mie University Board Committee for Animal Care and Use (#22-39-4, approved date was 3 Dec 2018). Six-week-old female KCASP1Tg and wild-type littermate mice were treated with anti-mouse RANKL monoclonal antibody (OYC1, Oriental Yeast, Kyoto, JAPAN) and Bonoteo^®^ (minodronate, Astellas, Tokyo, Japan). Anti-RANKL antibody is more effective than alendronate to suppress bone resorption. Minodronate is a third-generation bisphosphonate formulation, which has a stronger bone resorption inhibitory effect than that of alendronate. In general, alendronate, which is a world standard and often used in experiments, may be desirable as a control treatment group, but minodronate was selected in consideration of the strength of bone resorption inhibition. The treatment schedule was as follows: 5 mg/kg of OYC1 was administered subcutaneously every 4 days (*n* = 4) [22] and 1 mg/kg minodronate was administered subcutaneously every 28 days (*n* = 4). All mice were sacrificed at the age of 16 weeks.

### 4.2. Tissue Sampling

All mice were subjected to euthanasia with CO_2_ or pentobarbital. The whole blood was sampled by cardiac puncture and put in non-treated tubes. For serum collection, the blood samples were incubated at 4 °C for 16 h, and then centrifuged at 3000 rpm for 30 min at 4 °C. The collected serum was stored at −80 °C until examination. The femurs were sampled without excess muscle and patella tendon.

### 4.3. Real-Time Polymerase Chain Reaction (Real-Time PCR)

Total RNA was extracted from right femurs using tri reagent (Molecular Research Center, Cincinnati, OH, USA). After euthanasia, we rapidly sampled the right femur without muscle and patellar tendon, and the whole femur was immediately placed in tube containing tri reagent and minced. Femurs were thoroughly homogenized with BEAD CRUSHER *µ*T-01 (TAITEC, Saitama, Japan). The RNA was isolated with chloroform (nacalai tesque, Kyoto, Japan) and precipitated with isopropanol (nacalai tesque). The RNA concentration was measured using a NanoDrop Lite spectrophotometer (Thermo Fisher Scientific, Worsham, MA, USA), and 1 µg total RNA was converted to cDNA using a High-Capacity RNA-to-cDNA Kit (Applied Biosystems, Foster City, CA, USA). The Taqman Universal PCR Master Mix II with UNG (Applied Biosystems) was used to measure the mRNA expression of *Alpl* (Mm00475834_m1), *Col1a1* (Mm00801666_g1), *Bglap* (Mm03413826_mH), *Tnfsf11* (Mm00441906_m1), *Vegfa* (Mm00437306_m1), TNF-*α* (Mm00443258_m1), IL-1*α* (Mm00439620_m1), IL-1*β* (Mm01336189_m1), IL-6 (Mm00446190_m1), IL-17A (Mm00439618_m1), and IL-23*a* (Mm00518984_m1). *Gryceraldehyde-3-phosphate dehydrogenase* (*GAPDH*, Mm99999915_g1) was used as internal control. All probes were purchased from Applied Biosystems, and the amplification was performed in a LightCycler 96 System (Roche Diagnostics, Indianapolis, IN, USA). The cycling parameters were as follows: 50 °C for 120 s, 95 °C for 600 s, followed by 40 cycles of amplification at 95 °C for 15 s and 60 °C for 60 s.

### 4.4. ELISA

Tartrate-resistant acid phosphatase 5b (TRACP-5b, Immunodiagnostic Systems Limited, Bolden, UK) and osteocalcin (Elabscience, Wuhan, China) levels in mouse serum were measured by specific ELISA Kits. Experiment was performed in duplicate and the absorbance was measured using MULTISKAN JX (Thermo Fisher Scientific). Values were analyzed using Ascent Software for Multiskan Ascent (Thermo Fisher Scientific).

### 4.5. Measurement of BMD and Bone Architecture

The right femurs were fixed with 10% formalin neutral buffer solution, and then scanned by using a Scan Xmate-L090 micro-CT machine (µCT; Comscan Techno, Kanagawa, Japan) to measure the BMD and microarchitecture of the femurs. For BMD and trabecular bone assessment, the distal femur metaphysis was scanned at a voxel size of 10 µm and 16 µm and, for cortical bone assessment, the femur mid diaphysis and distal femur metaphysis were scanned at a voxel size of 10 µm. The acquisition of images was performed at a voltage of 75 kV with 100 µA. Image reconstruction was analyzed by using the TRI/3D-BON (Ratoc Systems Engineering, Tokyo, Japan). The following parameters were measured by *µ*CT: BMD, bone volume per tissue volume (BV/TV), trabecular thickness (Tb.Th), trabecular number (Tb.N), travecular separation (Tb.Sp), cortical bone volume per all bone volume (CV/AV), cortical bone section area (Cs), cortical thickness (Ct), and porosity (porosity was analyzed at distal femur metaphysis). All analyses were entrusted to Kureha Special Laboratory (Tokyo, Japan).

### 4.6. Histomorphometry

For histomorphometry, nonfixed and undecalcified left femurs were used for frozen section by the Kawamoto’s film method. The frozen blocks were cut into 5 µm slices by a Leica CM3050 S cryostat (Leica Biosystems, Wetzlar, Germany). Hematoxylin and eosin (HE) staining was performed in accordance with standard protocols. We measured cortical bone width at four randomly selected locations on HE of the overall femur by using analysis application hybrid cell count (Keyence, Osaka, Japan). Tartrate-resistant acid phosphatase (TRAP) was stained by TRAP/ALP Stain Kit (FUJIFILM WAKO, Osaka, Japan). Purified anti-mouse panendothelial cell antigen antibody (MECA-32, BioLegend, USA) was used for the detection of vascular endothelial cells. Microscopic images were captured in tiled array by a semi-automated BZ-X710 microscope (Keyence, Osaka, Japan) and analyzed by NIH ImageJ. The regions for the analysis of bone resorption and formation in histomorphometry were measured by photographing interest three section (x100) of the femoral diaphysis in each mouse. The following parameters were measured: number of osteoblasts per bone surface (Ob.N/BS), eroded surface per bone surface (ES/BS), number of osteoclasts per bone surface (Oc.N/BS), and surface of osteoclasts per bone surface (Oc.S/BS).

### 4.7. Statistical Analysis

A statistical analysis was performed by using PRISM software version 8 (GraphPad, San Diego, CA, USA). Two group comparisons were analyzed by Mann–Whitney test and three or more group comparisons were analyzed by Mann–Whitney test or ordinary one-way ANOVA. *p* values < 0.05 were considered indicative of statistically significant differences.

## 5. Conclusions

Persistent dermatitis caused osteoporosis. The impact of chronic dermatitis on osteoporosis has not been thoroughly investigated, and we need to study the association of dermatitis and osteoporosis.

## Figures and Tables

**Figure 1 ijms-21-03620-f001:**
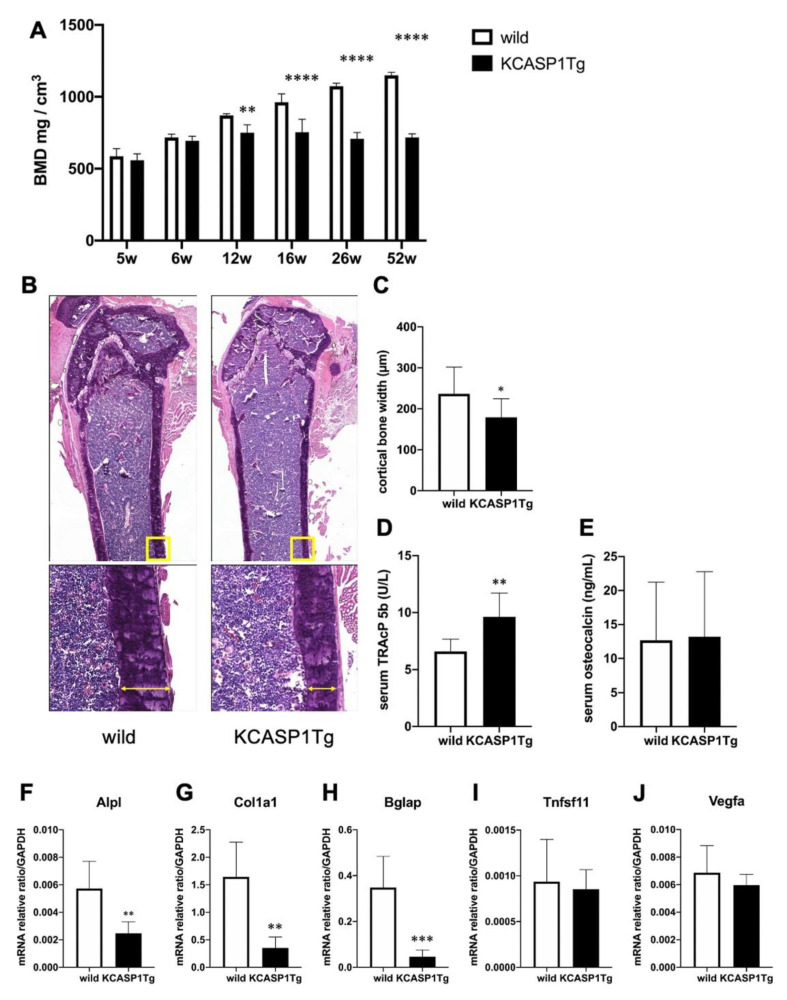
Bone mineral density (BMD) and expression of mRNAs associated to bone formation or destruction in the femur of KCASP1Tg and wild-type mice. BMD was scanned at the distal end of the right femur in KCASP1Tg and wild-type littermates of 5 to 52 weeks of age (*n* = 4 per group). In KCASP1Tg mice, BMD decreased significantly compared to wild-type littermates at 12 weeks of age (**A**). The cortical bone width was measured in HE (upper panel × 40, lower panel × 100). Four parts (*n* = 3 per group) were randomly selected and measured by using Analysis Application Hybrid cell count. The cortical width of the femur was significantly thinner in KCASP1Tg mouse, as determined by histological HE staining (**B,C**). Bone remodeling markers including TRACP-5b and osteocalcin were measured in the serum by a specific ELISA kit. The serum level of TRACP-5b is a sensitive marker of bone resorption, and was significantly increased in KCASP1Tg mice (**D**), but the biomarker of bone formation, serum osteocalcin, was unchanged ((**E**), *n* = 8 per group). The expression of relevant mRNAs in the right femur was quantified by real time PCR, and the values were standardized by using GAPDH (*n* = 6 per group). The expressions of genes involved in bone formation, such as alkaline phosphatase-(*Alpl*, (**F**)), collagen type I (*Col1a1*, (**G**)) and bone gamma-carboxyglutamate protein (*Bglap*, osteocalcin, (**H**)) were significantly decreased in KCASP1Tg mice. The mRNA levels of genes for osteoclast differentiation and activation, i.e., members of the tumor necrosis factor superfamily (*Tnfsf11*, *RANKL*, (**I**)) and vascular endothelial growth factor (*Vegfa*, (**J**)) were unchanged in the model mice. All data are expressed as mean ± SD. *; *p* < 0.05, **; *p* < 0.01, ***; *p* < 0.001, ****; *p* < 0.0001 between KCASP1Tg and wild-type mice by Mann–Whitney (except A) test and ordinary one-way ANOVA.

**Figure 2 ijms-21-03620-f002:**
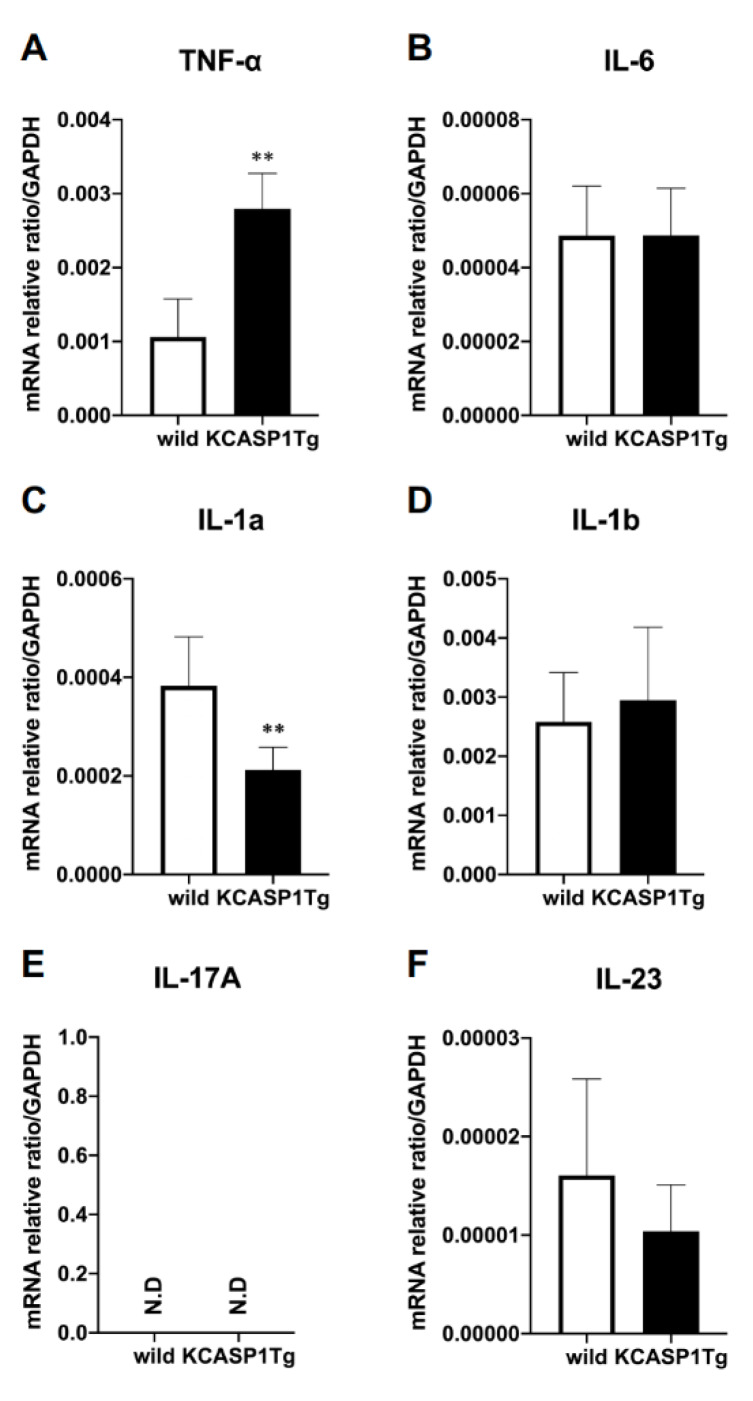
Inflammatory cytokine mRNA expression in the femur. The mRNA expression of inflammatory cytokines in the femurs was quantified using RT-PCR (*n* = 6 per group). The inflammatory cytokines TNF-*α*, IL-6, IL-1*α*, IL-1*β*, IL-17A, and IL-23*a* were measured and standardized using GAPDH. In the KCASP1Tg mice, TNF-α expression was significantly increased (**A**) and IL-1*α* expression decreased (**C**). There was no difference in the expression of IL-6, IL-1*β*, or IL-23*a* (**B,D,F**), and IL-17A was undetectable (**E**). All data are expressed as mean ± SD. N.D: not detected; **: *p* < 0.01 between KCASP1Tg and wild-type mice by Mann–Whitney test.

**Figure 3 ijms-21-03620-f003:**
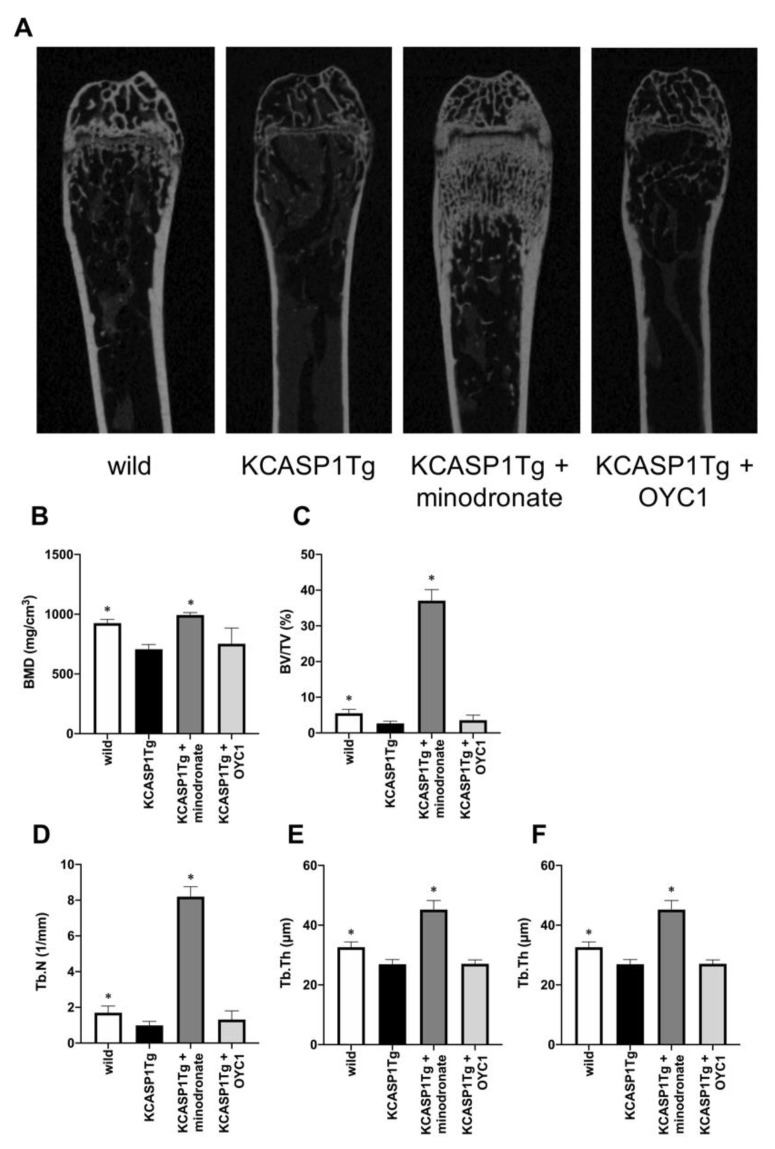
Analysis of the trabecular bone structure by *µ*CT. The trabecular bone structure was evaluated by *µ*CT in KCASP1Tg mice, treated with or without minodronate or OYC1 (*n* = 4 per each group) (**A**). BMD was decreased in KCASP1Tg mice and a marked augmentation of bone mass was observed in minodronate-treated KCASP1Tg mice (**B**). The bone volume per tissue volume (BV/TV) was reduced by 52% in KCASP1Tg mice compared to wild-type littermates and was significantly restored by minodronate (**C**). The trabecular number (Tb.N) was decreased in KCASP1Tg mice and was substantially recovered after treatment with minodronate, while OYC1 showed a moderate ameliorative effect (**D**). The distance between the trabeculae, the travecular separation (Tb.Sp), was increased (**E**), and the trabecular thickness (Tb.Th) was significantly decreased in KCASP1Tg mice (**F**). Minodronate induced a significant recovery. All data were expressed as mean ± SD. *; *p* < 0.05, compared to KCASP1Tg mice by Mann–Whitney test.

**Figure 4 ijms-21-03620-f004:**
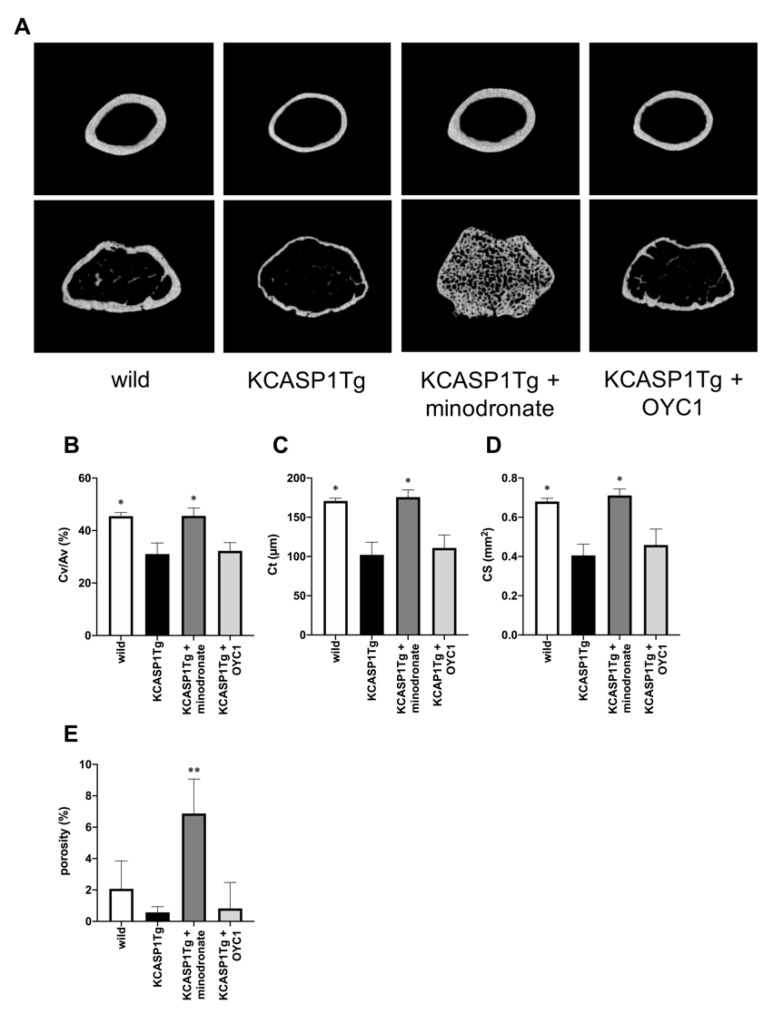
Analysis of cortical bone structure by µCT. The cortical bone structure of KCASP1Tg mice treated with or without minodronate or OYC1 was evaluated by µCT (*n* = 4 per each group, (**A**)). The cortical bone volume per all bone volume (Cv/Av) showed a 32% reduction in KCASP1Tg mice, but minodronate restored the wild-type level (**B**). The cortical thickness (Ct) and cortical bone section area (CS) at the femoral diaphysis were significantly decreased in KCASP1Tg mice compared to wild-type mice (**C,D**). In KCASP1Tg mice, the porosity was inconspicuous in the diaphyseal part, and decreased at the distal femur metaphysis; however, the difference was not significant (**E**). All data are expressed as mean ± SD. *; *p* < 0.05, **; *p* < 0.01 compared to KCASP1Tg mice by Mann–Whitney test.

**Figure 5 ijms-21-03620-f005:**
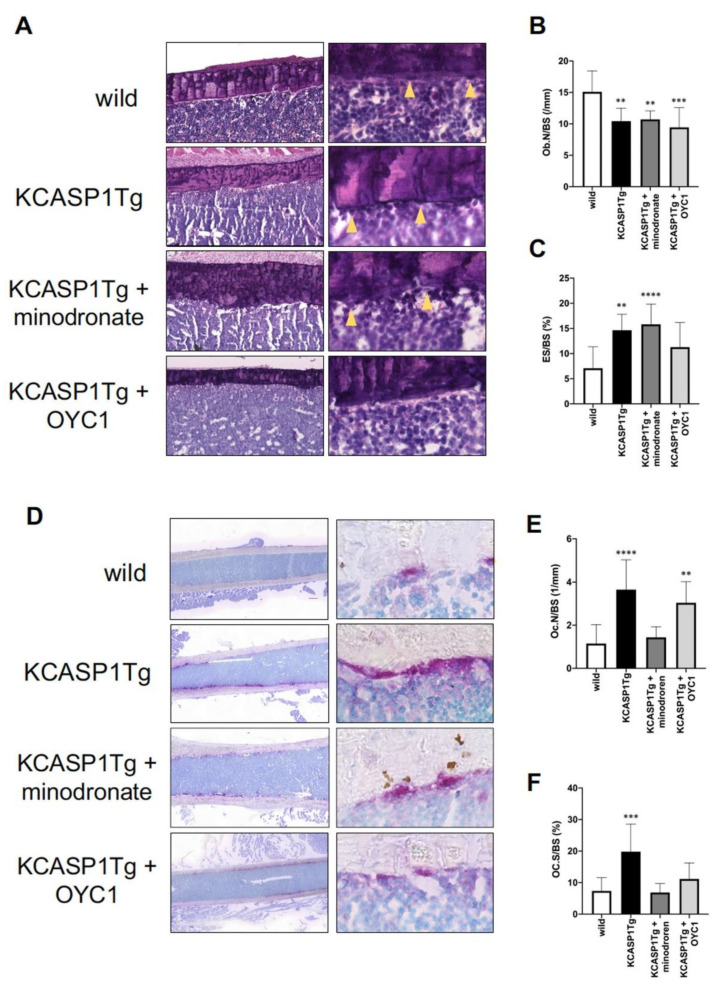
Histomorphometric evaluation by HE and TRAP staining. HE staining (left panel × 100 and right panel × 400) of the femoral diaphysis in wild-type, KCASP1Tg, and treated KCASP1Tg mice (**A**). The number of osteoblasts per bone surface (Ob.N/BS) was significantly decreased (**B**) and the eroded surface per bone surface (ES/BS) was increased in KCASP1Tg mice compared to wild-type littermates. ES/BS was not improved in both minodronate- and OYC1-treated KCASP1Tg mice (**C**). Ob.N/BS and ES/BS were quantified in three random parts of the HE section in each sample (×100, *n* = 3). TRAP staining (left panel scale bar is 300 µm, right panel × 400) of the femoral diaphysis in wild-type, KCASP1Tg, and treated KCASP1Tg mice (**D**). Osteoclast number per bone surface (Oc.N/BS, (**E**)) and osteoclast surface per bone surface (Oc.S/BS, (**F**)) were analyzed in three random parts of the TRAP section in each sample (×100, *n* = 3). Oc.N/BS and Oc.S/BS were significantly increased in KCASP1Tg mice compared to wild-type mice. Minodronate treatment reduced both Oc.N/BS and Oc.S/BS, whereas OYC1 ameliorated only Oc.S/BS (**E,F**). All data are expressed as mean ± SD. *; *p* < 0.05, **; *p* < 0.01, ***; *p* < 0.001, ****; *p* < 0.0001 compared to KCASP1Tg mice by Mann–Whitney test.

**Figure 6 ijms-21-03620-f006:**
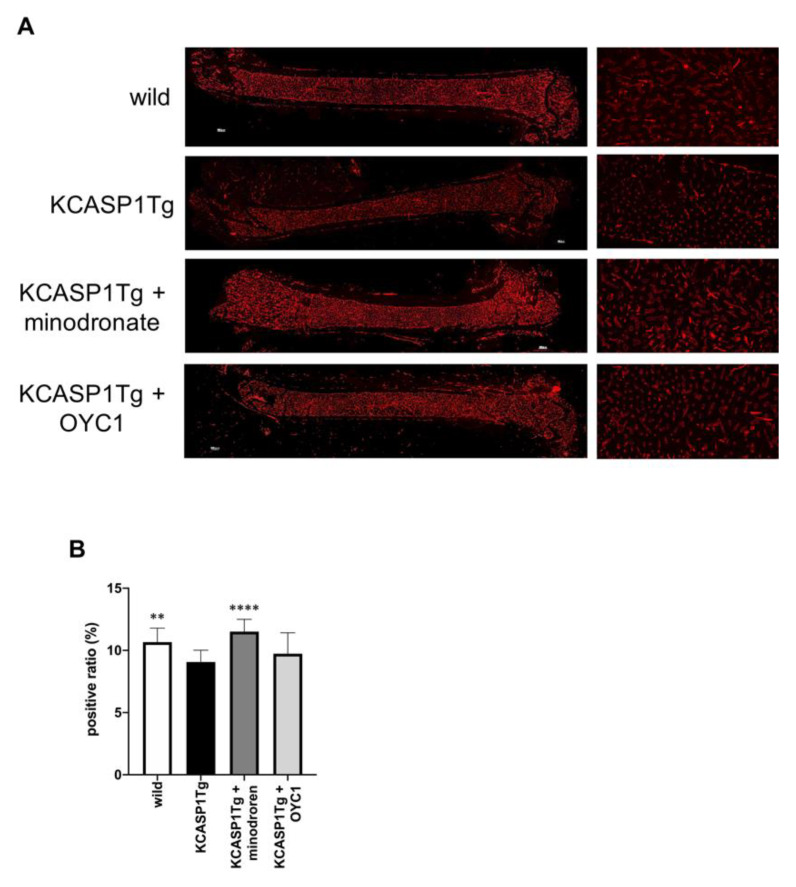
Blood flow evaluation in the femur by MECA-32 staining. In order to evaluate the blood flow in the femur, we performed fluorescent immunostaining with MECA-32, a vascular endothelial cell marker. In KCASP1Tg mice, the diameter of the blood vessels in the bone marrow was decreased (left panel scale bar is 300 µm, right panel × 100, (**A**)). We calculated the blood vessel area per bone marrow area (positive ratio) in three randomly selected locations (×100, *n* = 3 per group). The ratio was decreased in KCASP1Tg mice compared to wild-type littermates, while minodronate significantly improved the vascular area (**B**). **; *p* < 0.01, ****; *p* < 0.0001 compared to KCASP1Tg mice by Mann–Whitney test.

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
