# Peer review of "Inflammatory Skin-Derived Cytokines Accelerate Osteoporosis in Mice with Persistent Skin Inflammation"

_ijms, 2020, doi:10.3390/ijms21103620_

Round 1

Reviewer 1 Report

In this manuscript, the authors demonstrated that KCASP1Tg mice had osteoporosis. Interestingly, the osteoporosis can be at least partially rescued by minodronate treatment but not anti-RANKL antibody. However, this manuscript lacks mechanistic study and doesn't have any molecular insight why KCASP1Tg mice have osteoporosis.

Other concerns:

  1. It is unclear why KCASP1Tg mouse was used as a model in this study. Why this mouse but not other dermatitis mouse model. There are many much well-known animal models for psoriasis and AD. Have bone changes been observed in other models?
  2. This manuscript lacks logic and transition. For example, minodronate came out of blue. The authors didn't give any introduction about this drug and why to use this drug.
  3. A more detailed description about this mouse is necessary.
  4. Definition of Abbreviations are required in text.
  5. Description about many methods are missing. For example, a description how bone width was measured and how RNA was exacted from bone.
  6. Why was IL1a decreased? Any explanation?

Author Response

Responses to the comments of Reviewer #1

In this manuscript, the authors demonstrated that KCASP1Tg mice had osteoporosis. Interestingly, the osteoporosis can be at least partially rescued by minodronate treatment but not anti-RANKL antibody. However, this manuscript lacks mechanistic study and doesn't have any molecular insight why KCASP1Tg mice have osteoporosis.

 Response: Thank you for your comment. Indeed, our research does not elucidate the molecular relationship between dermatitis and osteoporosis. However, we could not find mouse study of secondary osteoporosis followed dermatitis. On the other hand, it has been reported that osteoporosis occurs clinically in patients with psoriasis vulgaris and psoriatic arthritis, and the presence of model mice is important in solving the mechanism. Our transgenic mouse which expresses specifically human caspase-1 in K14 reported in 2000 (Yamanaka, K., et al. (2000). The Journal of Immunology 165 (2): 997.) is a chronic and permanent dermatitis mouse model. At least, it was expected that osteoporosis could be caused by the release of inflammatory cytokines triggered by dermatitis, and we tried the treatment with anti-cytokine antibodies. However, it did not give sufficient results, probably because the dose and affinity of administrated antibodies were insufficient. We plan to study the pathogenic mechanism of dermatitis and osteoporosis by mating with various cytokine knock out mice, but we think that it is an important finding that osteoporosis may occur in dermatitis model mice. We appreciated you suggestion.

 Other concerns:

  1. It is unclear why KCASP1Tg mouse was used as a model in this study. Why this mouse but not other dermatitis mouse model. There are many much well-known animal models for psoriasis and AD. Have bone changes been observed in other models?

Response: We agree that lack of explanation to use KCASP1Tg mouse in this study. KCASP1Tg is a transgenic mice we have generated by ligating the human caspase-1 to the keratin 14 promoter, and it develops persistent dermatitis with IL-1beta (IL-1β) and various inflammatory cytokines secretion from inflamed skin. This has been added in the text. There are many well-known animal model for psoriasis and AD, but there is no report which showed secondary osteoporosis caused by skin inflammation. In fact, secondary osteoporosis related to skin inflammation was observed in our mouse model, despite the mouse model should have only skin inflammation. It could approach to reveal relationship between skin inflammation and osteoporosis. We have supplemented the explanation in the text. Thank you for your comments.

  2. This manuscript lacks logic and transition. For example, minodronate came out of blue. The authors didn't give any introduction about this drug and why to use this drug.

Response: Thank you for your suggestion. BMD in the femur of KCASP1Tg mice was significantly decreased, which was considered to be severe osteoporosis. We speculated the main cause was overproduced inflammatory cytokines from injured or inflamed skin. Therefore, the administration of anti-cytokine antibody had been tried. Other candidates for osteoporosis include anti-RANKL antibody and minodronate. Anti-RANKL antibody is more effective than alendronate to suppress bone resorption. Minodronate is a third-generation bisphosphonate formulation, which has a stronger bone resorption inhibitory effect that of alendronate. In general, alendronate, which is a world standard and often used in experiments, may be desirable as a control treatment group, but minodronate was selected in consideration of the strength of bone resorption inhibition. We have supplemented the explanation in the text. We appreciated your comments.

  3. A more detailed description about this mouse is necessary.

Response: Please refer to the response to your comment #1.

  4. Definition of Abbreviations are required in text.

Response: We have modified it.

  5. Description about many methods are missing. For example, a description how bone width was measured and how RNA was exacted from bone.

Response: Thank you for your advice. We have supplemented the description in the text.

  6. Why was IL1a decreased? Any explanation?

Response: Thank you for your kind question. In a previous report, we found that serum IL-1α level was increased in KCASP1Tg mice (Yamanaka, K., et al. (2000). The Journal of Immunology 165 (2): 997.), and we expected that mRNA expression of IL-1α in the femur was also increased, but the results were different. Horai et al. have reported the IL-1 receptor antagonist KO mice as rheumatoid arthritis model mice (Horai, R., et al. (2000). The Journal of experimental medicine 191(2): 313-320.). In this study, mRNA extracted from knee joint was measured; TNF-α and IL-1β were increased but IL-1α was decreased like our model mice. The reason for that was not stated, and we speculate that strong expression of TNF-α or IL-1β may suppress IL-1α, but there is no definite answer to this manifestation.

Reviewer 2 Report

This paper emphasizes importance of adequate treatment of skin condition fo not only skin lesions but also other organs. This is a nice paer.

Author Response

Responses to the comments of Reviewer #2

This paper emphasizes importance of adequate treatment of skin condition fo not only skin lesions but also other organs. This is a nice paer.

 Response: Thank you very much.

Reviewer 3 Report

This paper showed skin inflammation might accelerate irreversible osteoporosis, using KCASP1Tg mice. It would be more interesting if the author could mention the major cytokine which cause osteoporosis.

These mice have very severe skin inflammation, and various number of cytokines are extremely up-regulated. It is hard to say, this mice is a model of particular skin disease. Their new finding is that extreme skin inflammation would affect osteoporosis. The examination of osteoporosis of these mice are well described, although its mechanism is not enough clear.

Author Response

Responses to the comments of Reviewer #3

This paper showed skin inflammation might accelerate irreversible osteoporosis, using KCASP1Tg mice. It would be more interesting if the author could mention the major cytokine which cause osteoporosis.

 Response: Thank you very much.

Reviewer 4 Report

The study “Inflammatory skin-derived cytokines accelerate irreversible osteoporosis in mice

with persistent skin inflammation” by Kento Mizutani described the development of osteoporosis in a transgenic mous model with persistent skin inflammation.

Most of the data are convincing and the study is interesting.

My comments are the following:

Data in Figure 5: It is not really clear that ES/BS parameter emeliorates after treatment with OYC1, Panel C. As in the previous Figures the P values are calculated towards the wt mice by ANOVA. I think that in general, for testing the effect of treatment, it will be necessary to use a side by side comparison between the parameter measured in the untreated Tg mice vs the treated tg mice. The effect of OYC1 in this experiment does not seem so strong. Mann Whitney test could be more appropriate, or even Wilcoxon’s test (how many samples were tested?). It is possible that using these tests the data are not significant (at least for the effect described in Figure 5c).

The discussion is not really focused and is too long. The authors should mention that such data may need to be confirmed in other mouse models of psoriasis/atopic dermatitis.

Lane 30 31 Abstract: the statement is not correct, it may be … skin disease induced? And not induce?

Lane 72; I would not say that psoriasis is really ”intractable” as many biological therapies are now working in a satisfying manner (such as anti-IL-17 therapy).

Lane 265-268: there is a problem in this statement, I could not undestand what the author want to state.

Lane 284-285: there is a problem with the statement, I think is not correct and the significance of the phrase is difficult to be understood.

Author Response

Responses to the comments of Reviewer #4

The study “Inflammatory skin-derived cytokines accelerate irreversible osteoporosis in mice with persistent skin inflammation” by Kento Mizutani described the development of osteoporosis in a transgenic mous model with persistent skin inflammation. Most of the data are convincing and the study is interesting. My comments are the following:

  1. Data in Figure 5: It is not really clear that ES/BS parameter emeliorates after treatment with OYC1, Panel C. As in the previous Figures the P values are calculated towards the wt mice by ANOVA. I think that in general, for testing the effect of treatment, it will be necessary to use a side by side comparison between the parameter measured in the untreated Tg mice vs the treated tg mice. The effect of OYC1 in this experiment does not seem so strong. Mann Whitney test could be more appropriate, or even Wilcoxon’s test (how many samples were tested?). It is possible that using these tests the data are not significant (at least for the effect described in Figure 5c).

Response: Thank you very much for your suggestions. We apologize for the statistical error and retried using the Mann-Whitney test for KCASP1Tg in Figures 3, 4, 5 (three parts of random location was measured, and three mice were used in each group), and 6. As your suggestion, there was no significant difference in Figure 5C. We have revised the text. 

  1. The discussionis not really focused and is too long. The authors should mention that such data may need to be confirmed in other mouse models of psoriasis/atopic dermatitis.

 Response: We have deleted text and revised the discussion part. Currently, we have not seen a report in other model mouse models of psoriasis/atopic dermatitis. So we have supplemented it. Thank you. 

  1. Lane 30 31 Abstract: the statement is not correct, it may be …skin disease induced? And not induce?

Response: We apologize for the confusion. We would like to explain that there was no reported the efficacy of treatments against inflammatory skin disease in skin inflammation related secondary osteoporosis. The abstract has been modified. 

  1. Lane 72;I would not say that psoriasis is really ”intractable” as many biological therapies are now working in a satisfying manner (such as anti-IL-17 therapy).

Response: We agree your opinion and rephrased as “treatable”. 

  1. Lane 265-268: there is a problem in this statement, I could not undestand what the author want to state.

Response: We revised the text. 

  1. Lane 284-285: there is a problem with the statement, I think is not correct and the significance of the phrase is difficult to be understood.

Response: We agree and the sentence related to undetectable IL-17A was deleted.

Round 2

Reviewer 1 Report

The authors only responded some of the concerns. However, the authors failed to address the key concerns.